# Neural Regulation of Innate Immunity in Inflammatory Skin Diseases

**DOI:** 10.3390/ph16020246

**Published:** 2023-02-06

**Authors:** Xiaobao Huang, Fengxian Li, Fang Wang

**Affiliations:** 1Department of Dermatology, The First Affiliated Hospital, Sun Yat-sen University, Guangzhou 510080, China; 2Department of Anesthesiology, Zhujiang Hospital of Southern Medical University, Guangzhou 510282, China

**Keywords:** innate immunity, neuropeptide, neural regulation, atopic dermatitis, psoriasis

## Abstract

As the largest barrier organ of the body, the skin is highly innervated by peripheral sensory neurons. The major function of these sensory neurons is to transmit sensations of temperature, pain, and itch to elicit protective responses. Inflammatory skin diseases are triggered by the aberrant activation of immune responses. Recently, increasing evidence has shown that the skin peripheral nervous system also acts as a regulator of immune responses, particularly innate immunity, in various skin inflammatory processes. Meanwhile, immune cells in the skin can express receptors that respond to neuropeptides/neurotransmitters, leading to crosstalk between the immune system and nervous system. Herein, we highlight recent advances of such bidirectional neuroimmune interactions in certain inflammatory skin conditions.

## 1. Introduction

The immune system and the nervous system collaborate to protect the body from environmental threats. As the largest organ in the human body, the skin barrier is the first line to detect and combat various environmental stimuli such as toxins and pathogens [1]. In addition to the physical barrier function, the skin is also an important immune organ. The immune system is composed of two parts: the innate immune system and the adaptive immune system [2]. While the main purpose of innate immune responses is to immediately prevent the spread and movement of foreign pathogens throughout the body, the adaptive immune system builds the second line to defend against non-self elements. Although the skin is usually involved in infectious diseases, inflammatory disorders that are preceded by aberrant immune responses also occur and can significantly impair the quality of life of patients. The nervous system includes the peripheral nervous system (PNS) and central nervous system (CNS). The skin is highly innervated by the PNS, which includes sensory nerves and autonomic nerves [3]. Recently, seminal advances demonstrated that the nervous system interacting with the immune system—emerging as neuroimmunology—plays an important role in inflammatory processes. In this review, we aim to depict the classical themes and highlight updates of neuroimmunology underlying common inflammatory skin disorders.

## 2. Neuroimmune Crosstalk: A Novel Concept in Skin Inflammation

Inflammation is either caused by infections or aberrant immune responses. Regarding the characteristics of immune cells and mediators, immune responses are mainly classified into type 1, type 2, or type 3 [4]. Type 1 immunity, which is mediated by adaptive T helper (Th) type 1 cells, cytotoxic T cells, group 1 innate lymphoid cells (ILC1s), and natural killer cells, defends against intracellular pathogens and tumor cells through interferon (IFN)-γ production [2]. Adaptive Th2 cells, as well as innate basophils, eosinophils, ILC2s, and mast cells, mediate the type 2 immune responses by producing type 2 effector cytokines, such as interleukin (IL)-4, IL-5, and IL-13 [2]. The type 3 immune response protects against extracellular bacteria and fungi with the production of IL-17 and IL-22 from adaptive Th17 cells and gamma-delta T cells, as well as ILC3s and neutrophils [5,6]. Emerging evidence has demonstrated that immune cells in the skin are capable of releasing various cytokines that could directly act on skin nerve terminals [7,8]. Meanwhile, the skin PNS senses the stimulation and transmits the information to the CNS or nearby efferent neurons, leading to various neural symptoms such as itch and pain [9].

The skin comprises three major layers: epidermis, dermis, and subcutis [10]. The cutaneous immune cells mainly reside in the epidermis and dermis, where they are more likely to encounter environmental invaders. Antigen-presenting cells, including Langerhans cells and dermal dendritic cells (DCs), serve as part of the first line of defense against pathogens. Langerhans cells are often restricted to the epidermis, while dermal DCs stay deeper in the skin, and both of them are found in close proximity to nerve fibers [10,11]. Unlike mast cells that are usually skin-resident in the steady state, other myeloid cells, such as basophils, eosinophils, and neutrophils, are commonly found in the dermis in skin inflammatory conditions [12,13]. There are also ample T cells in the epidermis (mostly cytotoxic CD8^+^ T cells) and dermis (mostly CD4^+^ Th cells) [14].

Cutaneous sensory nerves, which derive from cell bodies in the dorsal root ganglia (DRG) and trigeminal ganglia, can be divided into three subtypes of nerve endings, which include Aβ, Aδ, and C nerve fibers based on the diameter and speed of transmission [15]. These sensory nerves are densely distributed throughout the dermis and epidermis. They mediate various sensations by encoding signals of pain, itch, temperature, pressure, position, and vibration [3]. Furthermore, these nerve fibers are anatomically close to the functional immune cells in the skin, providing the basis for cutaneous neuroimmune interactions [16,17] (Figure 1). It has been reported that pruritic diseases are closely related to immune cells [18]. For instance, mast cells have been implicated in various types of pruritus due to their capacity to release pruritogens [19]. However, neuropeptides and neurotransmitters released from activated neurons can act on the microvascular cells and resident mast cells to induce mast cell degranulation, which subsequently leads to a physiological reaction such as vasodilation and extravasation of plasma and leukocytes [20,21,22]. Therefore, the skin is a systemic organ that requires a neuroimmune interaction network to maintain its homeostasis.

## 3. Atopic Dermatitis (AD): A Paradigm of Itch Neuroimmune Mechanisms

AD is a chronic and relapsing skin disorder that is characterized by extensive itch and inflammatory skin lesions [23]. It is preceded by type 2 immune responses [24]. Although the role of the Th2 inflammatory axis, which leads to the immunoglobulin (Ig) E reactivity towards environmental allergens (e.g., pollen, house dust mites), is well established in its mechanisms [25], recent studies have demonstrated the importance of innate immune reactions. Moreover, the advances of itch mechanisms identify an important role of neuroimmune reactions in AD [26].

Histopathology examination has found that the skin lesions in patients with AD contain various innate immune cells that include mast cells, basophils, eosinophils, ILC2s, and DCs, leading to a possible role of innate immunity in AD [23]. Among these cells, mast cells represent a classic neuroimmune paradigm due to their close proximity to sensory nerves. Indeed, antigen recognition by IgE bound to FcεRI on mast cells results in IgE crosslinking and triggers the release of a variety of effector molecules, such as histamine, serotonin, proteases, and various cytokines [19]. These mediators are sufficient to activate sensory neurons and cause itch sensation [27]. Importantly, activated sensory neurons, in turn, could release neuropeptides or transmitters that act on mast cells to form a neuroimmune feedback loop. It has been demonstrated that mast cells have several receptors for neuropeptides, such as substance P (SP), calcitonin-gene-related peptide (CGRP), neuropeptide Y (NPY), and vasoactive intestinal peptide (VIP), expressed on their surface [28]. Decades ago, mast cells were discovered to degranulate in response to nerve growth factor (NGF) by the TrkA tyrosine receptor [29]. Interestingly, mast cells are also a source of NGF [30]. These data suggest that these mast cells and the sensory nervous system have a mutual interaction and thus form a skin–immune–nerve circuit.

More recently, Dong group found that a member of the Mas-related G protein-coupled receptor (MRGPR) family, MRGPRX2 (its mouse orthologue is MrgprB2), is expressed on the human mast cell surface [31,32]. Via binding to cationic molecules such as neuropeptides and host defense peptides, MRGPRX2 encourages mast cells to degranulate and release tryptase, rather than histamine [31]. Since tryptase is also a pruritogen that could evoke itch scratching, these new findings indicate the presence of heterogeneous neuroimmune pathways underlying innate immunity. Despite this, studies on MRGPRX2 in the context AD are still limited. Although mast cells are increased in skin lesions and show signs of degranulation in AD [33], the mechanisms of their activation modes are surprisingly ill-defined. Interestingly, nerve fibers in AD skin lesions show greater positive staining for SP [34], provoking the hypothesis of an intense neural–mast cell communication in AD. Due to the expression of MRGPRX2 in connective tissue mast cells of the skin [35], Wang et al. proposed a possible role of MRGPRX2 in AD: while MRGPRX2-activated mast cells release inflammatory cellular contents that act on sensory nerves, neuropeptides from sensory fibers promote mast cell activity via MRGPRX2 in the skin. In an allergic skin inflammation model induced by house dust mites with cysteine protease activity, Serhan et al. found that transient receptor potential vanilloid 1+ neurons were able to promote mast cell activation via the release of SP, which causes mast cell degranulation by binding to MrgprB2 [36]. Moreover, patients with AD often have *Staphylococcus aureus* isolated from their skin [37]. *Staphylococcus aureus* can also directly activate sensory neurons to release SP [38]. Given the activation of MrgprB2 on mast cells is essential for host defense against *Staphylococcus aureus* [39], these data indicate that mast cells and sensory nerves form an intrinsic and cyclical relationship [40].

Basophils share multiple similar features with tissue-resident mast cells. Increasing discoveries have shown that basophil recruitment to tissues enables them to exhibit unique functions in skin inflammatory disorders. Similar to mast cells, basophils respond rapidly to the IgE-mediated activation by degranulating a variety of pre-stored effector molecules such as histamine, and later leukotrienes, IL-4, and IL-13 [41]. Interestingly, Oetjen et al. discovered that the orchestration of sensory neurons and type 2 cytokines could be the key mechanism of chronic pruritus [7]. They demonstrated that the type 2 cytokines IL-4 and IL-13 could directly activate sensory neurons in both mice and humans. Therefore, basophils may play an important role in chronic itch due to their capacity to release IL-4 and IL-13. Apart from chronic itch, we found that basophils also contribute to acute itch flares in a mast-cell-independent manner in the context of AD [42]. Specifically, the basophil-associated leukotriene C4 and its receptor CysLTR2 on sensory neurons are critically required to transmit this form of IgE-mediated itch. Similar to mast cells, functional MRGPRX2 is also significantly expressed by basophils, and the upregulation of MRGPRX2 is associated with basophil degranulation and CD63 expression [43]. Collectively, these data indicate that basophils may be an important modulator of AD-associated itch.

ILCs are immune cells that belong to the lymphoid lineage but do not express antigen-specific T cell receptors. ILC2s serve crucial functions in various different tissues, but are especially enriched in barrier tissues, such as the lungs, gut, and skin. Due to the specific receptor expression on ILC2s, epithelial-cell-derived cytokines such as IL-25, IL-33, and thymic stromal lymphopoietin (TSLP) are sufficient to activate ILC2s [44,45]. Notably, it has been demonstrated that human AD skin has elevated expression of IL-33, IL-25, and TSLP [46,47,48], indicating a role of ILC2s in AD. Indeed, Kim et al. showed that ILC2s significantly accumulate in skin lesions of patients with AD [49]. Furthermore, those ILC2s stay in close proximity to basophils in the inflammatory dermis. Utilizing an AD murine model mediated by topical MC903 (a calcipotriol analog), authors have demonstrated that basophil-derived IL-4 plays a significant role in promoting ILC2s in the setting of AD-like disease. Following activation, skin ILC2s could constitutively produce IL-5 and IL-13, leading to AD-like pathology and Th2 responses [48,50,51,52]. Collectively, these studies demonstrate that skin ILC2s promote type 2 skin inflammation and coordinately interact with adaptive cells in the process of AD.

The crosstalk between ILC2s and the PNS has been demonstrated in the context of allergic lung inflammation. Cardoso et al. found that murine mucosal ILC2s selectively express neuromedin U (NMU) receptor 1 and co-localize with cholinergic neurons that express NMU [53,54]. Following stimulation with IL-25 and IL-33, NMU induces ILC2 proliferation and activation to produce type 2 cytokines and promote lung inflammation. Nagashima et al. employed a lung helminth infection model and discovered that α-CGRP limits ILC2 proliferation and IL-13 production by antagonizing the action of NMU and IL-33 [55]. However, the production of IL-5 in ILC2s is promoted in the same context. The Locksley group showed that mucosal ILC2s also have mRNA expression of both VIP receptor type 1 and type 2 [56]. Moreover, ILC2s isolated from murine intestine had an elevated IL-5 release with the addition of VIP or activation of VIP receptor type 2 [56], providing another neuroimmune pathway of ILC2 regulation.

Due to the tight bond between ILC2s and basophils, Inclan-Rico et al. re-explored the role of ILC2 in the murine model of lung *Nippostrongylus brasiliensis* infection [57]. They found that helminth-induced ILC2 responses are exaggerated in the setting of basophil depletion, resulting in increased inflammation and diminished lung function. More importantly, a neuropeptide, neuromedin B (NMB), has similar effects on ILC2s to basophil deficiency. Authors have further shown that the role of basophils in this setting is to enhance ILC2s to increase the expression of the NMB receptor. Although the major source of NMB in this setting remains undefined, these findings suggest that basophils have the capacity to prime ILC2s to respond to neuropeptides, and that NMB might be a potent inhibitor of type 2 inflammation. To date, neuroimmune studies based on ILC2s in the context of AD remain quite few. However, given those previous findings on neuropeptides and ILC2s in other settings of type 2 inflammation (Figure 2), we are optimistic that novel neuroimmune mechanisms of AD will be disclosed in the near future.

DCs are a subset of innate immune cells and have unique functions of antigen uptake and presentation, which are essential for the production of allergen-specific Igs. The distribution of DC subsets depends on the different phases of inflammation [58]. In skin lesions from AD patients, inflammatory epidermal DCs (positive for CD11c and CD206) were markedly shown in central areas of the spongiotic epidermis [59]. These DCs are believed to induce T cell responses and provide a potential target for therapeutics. Interestingly, a recent study conducted by Perner et al. employed an allergen recognition model in which papain was directly injected into murine skin [60]. They further found that SP and its receptor MrgprA1 are critically required for the migration of CD301b^+^ DCs from the skin to the draining lymph node where they initiate Th2 cell differentiation. In human studies, NPY has been shown to induce the migration of human monocyte-derived immature DCs through its engagement of the NPY Y1 receptor in vitro, resulting in Th2 polarization [61]. Unlike NPY, the effects of VIP on DCs are multifaceted. Immature DCs following the treatment of VIP, both in vivo and in vitro, have increased CD86 expression to promote CD4^+^ T cell proliferation and to exhibit a Th2 phenotype [62]. In contrast, in the setting of lipopolysaccharide stimulation, VIP reduces CD86 and CD80 expression on DCs and thus leads to an inhibitory effect on T cell proliferation [62]. Brain natriuretic peptide transmits the itch signal in the nervous system. In AD, brain natriuretic peptide was found to be released from sensory nerves that are activated by IL-31 [63]. In turn, brain natriuretic peptide stimulates DCs to release inflammatory factors, which contributes to a positive feedback loop of neuroinflammation [63,64].

Langerhans cells are epidermal-resident DCs and play a key role in the initiation of cutaneous immune responses. Due to the close association between Langerhans cells and nerves, several studies have investigated the neural effects on Langerhans cells. In a murine model of ovalbumin-specific presentation, Ding et al. showed that CGRP enhances Langerhans cell function and Th2 responses by increasing IL-4, CCL17, and CCL22 production and decreasing IFN-γ production [65]. These studies indicate that neuropeptides are regulators in the process of allergen presentation and may open new avenues to help regulate the pathologic process of AD.

## 4. Psoriasis: Neurogenic Skin Inflammation

Psoriasis is a chronic inflammatory skin disease shaped by genetics, environmental factors, and psychological stress [66]. Both clinical and experimental studies have suggested an involvement of neurogenic components in its pathogenesis [67]. The aberrant type 1/17 immune responses represent a predominant mechanism in psoriasis due to the abundant release of type 1/17 cytokines such as IFN-γ, IL-17A, IL-23, and tumor necrosis factor (TNF)-α [68]. These cytokines lead to keratinocyte proliferation as well as sustained skin inflammation.

Chronic itch is a symptom occurring not only in patients with AD, but also in patients with psoriasis. Although mast cells may not be key to AD-related itch, they partially contribute to the itch mechanisms in psoriasis [67]. Histopathologic examination of skin lesions in patients with psoriasis revealed that, compared with non-pruritic skin tissues, the pruritic areas had more mast cells and degranulated mast cells [69]. Additionally, a rich innervation of nerves and an increase in SP-, CGRP-, and VIP-positive nerve fibers were also shown in the psoriatic skin [10]. In addition, there is a greater frequency of morphological contacts between neurofilament+ nerves and tryptase+ mast cells in psoriatic lesions than in non-lesional skin or normal-looking skin, suggesting a morphologic basis for mast cell–neural interactions in this disease [70]. Notwithstanding this, the effects of antihistamines on psoriasis are often limited, provoking the hypothesis that other inflammatory mediators are involved in this process. Indeed, itch-associated cytokines or pruritogens such as IL-31, TSLP, and tryptase have been found to be elevated in skin lesions at the transcriptomic level [69,71,72], indicating a complex of itch mechanisms in psoriasis.

Neurogenic inflammation is believed to play an important role in psoriasis. The direct evidence is that SP-immunoreactive fibers and SP receptors (neurokinin 1 receptor (NK1R) and MRGPRX2) in the skin of psoriatic patients with itch have a higher expression than those in patients without itch [69,73]. Meanwhile, elevated plasma CGRP levels were observed in psoriatic patients [74]. The receptor of CGRP is also detected in psoriatic skin lesions [75]. It is well established that SP and CGRP are sufficient to induce mast cells to release various cytokines such as IL-1β and TNF-α, which can attract neutrophils to accumulate in the skin [69,76]. Neutrophils can produce antimicrobial peptides such as α-defensins, thus leading to persistent inflammation [77]. Smith et al. conducted an interesting experiment by intradermally injecting CGRP into normal human skin and found a significant infiltration of neutrophils [78]. In addition to SP and CGRP, galanin is another bioactive neuropeptide. One of its receptors, galanin-R3, is expressed on the vascular endothelium. In a murine model of psoriasis-like disease mediated by imiquimod, the lack of galanin-R3 led to a less severe disease phenotype that included delayed neo-vascularization, reduced neutrophil infiltration, and significantly lower levels of proinflammatory cytokines in contrast to the controls [79]. An interesting phenomenon in psoriasis is called the Koebner phenomenon, in which a trauma or wound can cause psoriatic lesions in normal-looking skin. However, its mechanisms remain unclear. NGF appears to be an initial neuropeptide in this process since it is capable of causing the release of other neuropeptides, including SP and CGRP [67]. Meanwhile, elevated expression of NGF and its receptor TrkA was also exhibited in psoriatic lesions [80]. Taken together, these findings suggest that the nervous system likely contributes to skin inflammation in psoriasis (Figure 3).

The direct evidence that sensory nerves, particularly nociceptors, are involved in the pathogenesis of psoriasis-like disease was published in 2014 [81]. In this study, by employing the imiquimod-induced psoriasis murine model, the authors found that the activated transient receptor potential V1^+^ nerves innervating the skin are required to promote IL-23 production in dermal DCs, which, in turn, can stimulate dermal gamma-delta T cells to secrete IL-17A, IL-17F, and IL-22, resulting in the recruitment of more neutrophils to the skin and excessive keratinocyte proliferation. These findings shed light on how the nervous system affects the reaction of local innate immune cells and thereby contribute to the mechanisms of psoriasis.

ILC3s are defined by their capacity to produce IL-17A and/or IL-22. Similar to Th17 cells, ILC3s depend on the transcription factor RORγt for their development and function [82]. ILC3s are considered to contribute to the pathogenesis of psoriasis [82]. Natural cytotoxicity receptor-positive ILC3s have been found to be increased in psoriatic lesions [83]. Following stimulation with IL-23 and IL-1β, these ILC3s isolated from psoriatic skin lesions can produce IL-22 ex vivo. Although studies on the effects of neuropeptides on ILC3 regulation in the context of psoriasis remain few, some studies have explored neural–ILC3 interactions in the gut. A recent study demonstrated that intestinal ILC3 responses depend on the food-induced expression of the neuropeptide VIP [84]. Intestinal ILC3 had a high expression of VIP receptor 2. Enteric neuron-derived VIP binds to this receptor on ILC3s and causes the production of IL-22, which is critical for the maintenance of homeostasis in the intestinal tract. The deficiency in signaling through VIP receptor 2 increased susceptibility to inflammation-induced gut injury by reducing IL-22 production from ILC3. In contrast, Talbot et al. discovered that VIPergic neurons reduced IL-22 production by CCR6^+^ILC3s through VIP receptor 2 [85]. They found that the presence of commensal microorganisms upregulates ILC3s, which, in turn, can enhance IL-22 production, and this phenomenon is inhibited by the engagement of VIP receptor 2. Moreover, Yu et al. have discovered that VIP promotes ILC3 recruitment to the intestine through VIP receptor 1 [86]. Although discrepancy exists among different studies, these findings establish scientific clues for future research on the neural–ILC3 axis in psoriasis (Figure 3).

The complex network of DCs in the skin tissue is composed of Langerhans cells, conventional bone-marrow-derived dermal DCs, plasmacytoid DCs, and inflammatory DCs. Via the activation of Toll-like receptors, Langerhans cells purified from normal healthy skin produce high levels of IL-23, which is crucial in the pathogenesis of psoriasis [87,88]. Meanwhile, activated conventional DCs secrete a range of inflammatory cytokines such as IL-12 and IL-23 [89]. Intriguingly, it has been shown that CGRP is required for the infiltration of DCs and T cells into the skin in a psoriasiform murine model [90]. The authors utilized a small molecule to inhibit CGRP and found a reduction in the CD4^+^ T cell number as well as skin acanthosis. Although α-melanocyte-stimulating hormone has been found to stimulate melanogenesis, which is responsible for the pigmentation of the hair and skin, a study by Matteo et al. found that α-melanocyte-stimulating hormone, binding to MC1R, is sufficient to induce tolerogenic DCs and leads to the proliferation of regulatory T cells and inhibition of Th17 activities [91]. Another study by Ding et al. found that VIP could promote the process of antigen presentation and the secretion of IL-17A and IL-6 from Langerhans cells, indicating a role of VIP in psoriasis-related mechanisms [92].

## 5. Other Inflammatory Skin Diseases: Neuroimmune Responses Likely Involved

In addition to AD and psoriasis, other cutaneous disorders that may involve neuroimmune interactions at least include chronic spontaneous urticaria (CSU) and bullous pemphigoid. Previously published studies showed that the intradermal injection of SP and VIP induces exaggerated responses in the skin of patients with CSU [93,94]. Moreover, the SP receptor MRGPRX2 has elevated expression levels on mast cells in the context of CSU [95]. Additionally, Zheng et al. found that circulating SP^+^ and NK1R^+^ basophils were markedly elevated in CSU in comparison with healthy controls, and SP is capable of causing basophil degranulation and tissue accumulation [96]. In patients with bullous pemphigoid, both NK1R^+^ cells and SP^+^ cells were exhibited in the skin [97]. More importantly, both NK1R^+^ cells and SP^+^ cells have a positive correlation with the itch severity. Given that most NK1R^+^ cells are identified as eosinophils, it would be interesting to explore the neuro–eosinophil interactions in bullous pemphigoid in future studies.

## 6. Conclusions

Emerging evidence has shown that the neuroimmune crosstalk plays a significant role in the regulation of skin inflammation. We highlighted the updates of the neuroimmune interactions underlying AD and psoriasis (Table 1). Although several groups including us have identified novel neuroimmune mechanisms in skin inflammation, a lot of questions remain unclear. For instance, the heterogenous functions of neuropeptides on various immune cells are not fully investigated. Due to the small amount of production of neuron-derived molecules and their rapid action within nerve ending microenvironments, the detection and quantification of these molecules in skin tissues are technically challenging. Beyond neurons, immune cells and keratinocytes could also be a source of neuropeptides, indicating that an autocrine or paracrine mechanism might contribute to the immune–nerve circuit. Therefore, it is necessary to clarify the cellular source and function of various neuropeptides in the skin and other peripheral organs to deeply explore the complex of neuroimmune responses.

In recent decades, an increasing number of novel targeted therapies, especially biologics, have been brought into clinical use for inflammatory skin diseases. Here, we summarize the therapeutic targets and medications that are associated with neuroimmune interactions for skin inflammatory disorders (Table 2). In the future, with more neuroimmune interactions unveiled, more effective treatment options will be introduced to skin inflammatory disorders.

## Figures and Tables

**Figure 1 pharmaceuticals-16-00246-f001:**
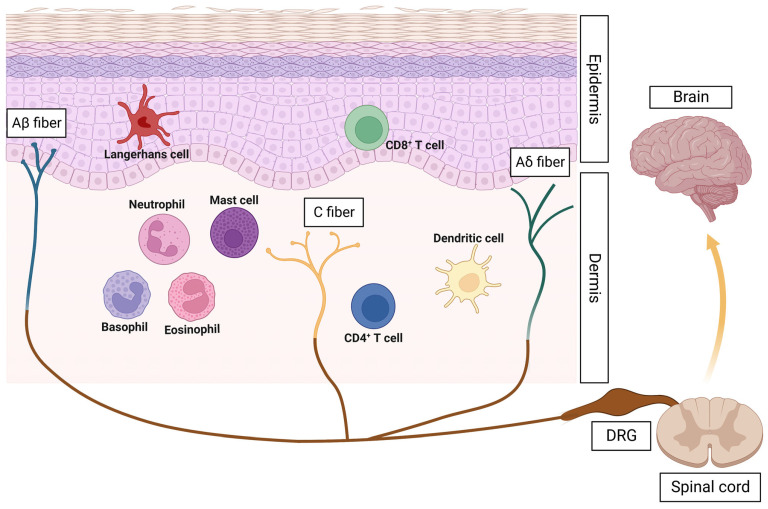
Involvement of immune cells and sensory nervous system in the skin. Three components are shown: the stratified epidermis and dermis; representative innate and adaptive immune cells; the sensory nerves that are derived from dorsal root ganglia (DRG), which have afferent endings anatomically close to functional immune cells. Figure created with Biorender.

**Figure 2 pharmaceuticals-16-00246-f002:**
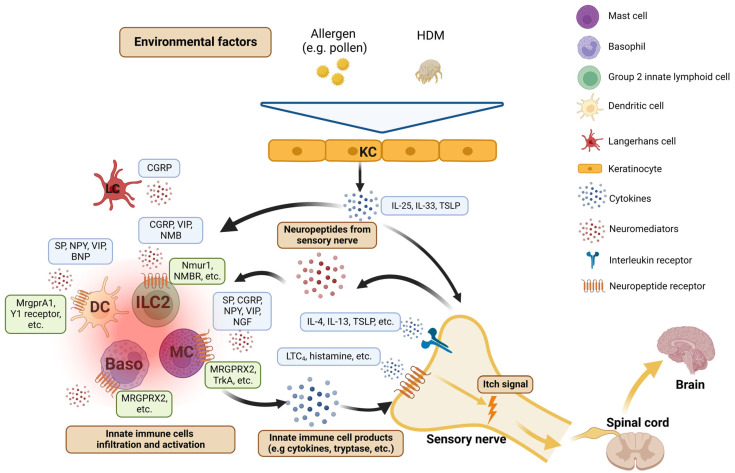
The skin–immune–nerve circuit in AD and type 2 inflammation. Environmental stimuli cause keratinocytes to release epithelial cytokines that trigger a type 2 inflammatory axis in AD. Either non-self elements or effector molecules (e.g., histamine, serotonin, proteases, cytokines) could directly activate sensory neurons to provoke itch sensation. Activated sensory neurons, in turn, release neuropeptides or transmitters to the skin, thereby aggravating inflammation through the modulation of immune responses. Baso, basophil; CGRP, calcitonin-gene-related peptide; DC, dendritic cell; HDM, house dust mites; IL, interleukin; ILC2, group 2 innate lymphoid cell; KC, keratinocyte; LC, Langerhans cell; LTC4, leukotriene C4; MC, mast cell; MRGPR, Mas-related G protein-coupled receptor; NMB, neuromedin B; NMBR, neuromedin B receptor; Nmur, neuromedin U receptor; NPY, neuropeptide Y; SP, substance P; TSLP, thymic stromal lymphopoietin; VIP, vasoactive intestinal peptide. Figure created with Biorender.

**Figure 3 pharmaceuticals-16-00246-f003:**
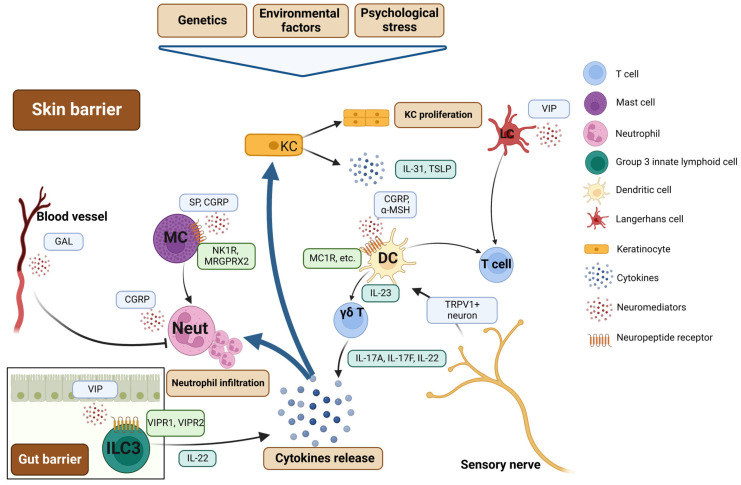
Connection of neural mediators and innate immune responses in psoriasis and type 1/17 inflammation. Genetics, environmental factors, and psychological stress shape keratinocytes in the context of psoriasis. Type 1/17 cytokines lead to keratinocyte proliferation and sustained skin inflammation. Neuropeptides, mainly from nerves, are sufficient to activate the innate immune responses either in the skin or gut, leading to immune cell recruitment, cytokine release, and persistent inflammation. α-MSH, α-melanocyte-stimulating hormone; CGRP, calcitonin-gene-related peptide; DC, dendritic cell; GAL, Galanin; IL, interleukin; ILC3, group 3 innate lymphoid cell; KC, keratinocyte; LC, Langerhans cell; MC, mast cell; MC1R, melanocortin 1 receptor; MRGPR, Mas-related G protein-coupled receptor; Neut, neutrophil; NK1R, neurokinin 1 receptor; SP, substance P; TSLP, thymic stromal lymphopoietin; VIP, vasoactive intestinal peptide; VIPR, VIP receptor. Figure created with Biorender.

**Table 1 pharmaceuticals-16-00246-t001:** Main neuroimmune mechanisms in atopic dermatitis and psoriasis.

	Atopic Dermatitis or Type 2 Inflammation	Psoriasis or Type 1/17 Inflammation
Predominant inflammatory factors	IL-4, IL-13, IL-31, IL-25, IL-33, TSLP	IFN-γ, IL-17A, IL-22, IL-23, TNF-α
Neuropeptide mechanism		
Substance P	Promotes mast cell activation and degranulation; promotes migration of CD301b+ DCs	Promotes mast cell degranulation
Calcitonin-gene-related peptide	Promotes mast cell degranulation; limits ILC2 proliferation and IL-13 production; enhances LC function	Promotes mast cell degranulation; promotes infiltration of DCs and T cells
Vasoactive intestinal peptide	Promotes mast cell degranulation; promotes IL-5 release from ILC2s; induces Th2 polarization	Promotes mast cell degranulation; promotes ILC3 recruitment; influences IL-22 production from ILC3s; enhances LC function
Brain natriuretic peptide	Stimulates inflammatory factors from DCs	Unknown
Neuropeptide Y	Promotes mast cell degranulation; induces migration of human immature DCs	Unknown
Neuromedin B	Limits type 2 inflammation	Unknown
Neuromedin U	Induces ILC2 proliferation and activation	Unknown
Nerve growth factor	Promotes mast cell degranulation	Enhances basophil function
Galanin	Unknown	Induces neo-vascularization, neutrophil infiltration, and cytokine release
α-Melanocyte-stimulating hormone	Unknown	Induces tolerogenic DCs; leads to Treg proliferation; inhibits Th17 activities

Abbreviations: DC, dendritic cell; IFN-γ, interferon-γ; IL, interleukin; ILC, innate lymphoid cell; LC, Langerhans cell; TNF-α, tumor necrosis factor α; Treg, regulatory T cells; TSLP, thymic stromal lymphopoietin.

**Table 2 pharmaceuticals-16-00246-t002:** Emerging therapeutic agents associated with neuroimmune interactions.

Target	Therapeutic Agents	Mode	Indications	References/NCT Number
NK-1R	Aprepitant	Antagonist	AD; chronic prurigo	[98,99]
	Serlopitant	Antagonist	AD; psoriasis; PN; chronic refractory pruritus	[100,101,102,103]; NCT02975206; NCT03343639; NCT03546816; NCT01951274
	Tradipitant	Antagonist	AD	[104,105]; NCT02004041; NCT02651714
TrkA	Pegcantratinib	Inhibitor	Psoriasis	[106]; NCT03448081
KOR/MOR	Nalbuphine	Agonist of KOR/antagonist of MOR	Chronic prurigo	NCT02174419; NCT02174432
TRPV1	Asivatrep	Antagonist	AD	[107,108]
IL-13	Tralokinumab	mAb	AD	[109]; NCT03363854
	Lebrikizumab	mAb	AD	[110]; NCT04146363; NCT04178967; NCT04392154; NCT04250350; NCT04250337
IL-33	Etokimab	mAb	AD	[111]; NCT03533751
TSLP	Tezepelumab	mAb	AD	[112]; NCT02525094; NCT03809663
IL-4Rα	Dupilumab	mAb	AD; PN; chronic pruritus	[113,114,115,116,117]
IL-31RA	Nemolizumab	mAb	AD; PN	[118,119,120,121]

Abbreviations: AD, atopic dermatitis; KOR, kappa opioid receptor; mAb, monoclonal antibody; MOR, Mu opioid receptor; NCT, National Clinical Trial; NK-1R, neurokinin 1 receptor; PN, prurigo nodularis; TRP, transient receptor potential; TSLP, thymic stromal lymphopoietin.

## Data Availability

Not applicable.

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
