# Peer review of "Neural Regulation of Innate Immunity in Inflammatory Skin Diseases"

_pharmaceuticals, 2023, doi:10.3390/ph16020246_

Round 1

Reviewer 1 Report

Huang et al. have performed an interesting review on the neural regulation of innate immunity in inflammatory skin diseases. There are several things that can be improved.

In the abstract - line 18 - the authors should mention that they discuss about inflammatory skin conditions. 

There are no bibliographic sources in the Introduction section. The authors should revise it.

Neuroimmune crosstalk: a novel concept in skin inflammation Section should be improved. More data could be added. There are several recent articles that have addressed this topic doi: 10.1016/j.jid.2021.10.006, doi: 10.3389/fimmu.2022.1003970

Lines 187-193 – the authors should add bibliographic sources.

The titles of the manuscript sections should reflect their content. The titles of Sections 3 and 4 refer only to the link between the disease presented and inflammation and do not mention the nervous component.

Several important reviews have not been included (doi.org/10.1111/exd.14071, doi.org/10.1016/j.jaci.2022.03.010)

The conclusions should derive from the data presented and discussed in the manuscript and represent the personal contribution of the authors. New data accompanied by bibliographic references should not be included in the Conclusions section.

Reviewer 2 Report

This manuscript, review type, written by Dr Huang, with the title of “Neural Regulation of Innate Immunity in Inflammatory Skin Diseases” described the neuro-immune interaction is two diseases, atopic dermatitis and psoriasis. The manuscript is well written, and it is quick to read. To improve the manuscript, the authors could make more tables and figures that summarize that pathological mechanisms, and/or add other skin diseases.

Line 42, regarding “classified into type 1, type 2, and/or type 3”. Could you please describe the 3 types of immune responses?

Lines 39-53. Could you please show the histological characteristics of the skin, including the presence of immune system and nerve fibers? Could you please show the physiological neuroimmune interactions?

Could you please try to avoid using many abbreviations?

Could you please add an explanation of the different pathogenic mechanisms shown in Figure 1, in the text of Figure 1? (Figure 1 has only the title, and it is difficult to understand all mechanisms). Or the text of Figure 1 are the lines 174-185?

Could you please make a table summarizing the pathogenic mechanisms for both diseases (AD, and psoriasis)?

Could you please add a table of possible therapeutic agents? (Please refer to https://doi.org/10.3389/fmed.2021.627985).

Could you please refer to this publication? There are very nice figures, and data. https://onlinelibrary.wiley.com/doi/10.1002/eji.201848027

Round 2

Reviewer 1 Report

The manuscript has been significantly improved.